# Comparative Performance of Digital PCR and Real-Time RT-PCR in Respiratory Virus Diagnostics

**DOI:** 10.3390/v17091259

**Published:** 2025-09-18

**Authors:** Irene Bianconi, Giovanna Viviana Pellecchia, Elisabetta Maria Incrocci, Fabio Vittadello, Maira Nicoletti, Elisabetta Pagani

**Affiliations:** 1Laboratory of Microbiology and Virology, Hospital of Bolzano (SABES-ASDAA), Teaching Hospital of Paracelsus Medical University (PMU), 39100 Bolzano-Bozen, BZ, Italy; 2Scuola Provinciale Superiore di Sanità Claudiana, 39100 Bolzano-Bozen, BZ, Italy; 3Explora-Research and Statistical Analysis, 35010 Padova, PD, Italy

**Keywords:** digital PCR, Real-Time RT-PCR, respiratory viruses, tripledemic, virus diagnostics

## Abstract

*Background*: Respiratory viral infections pose a major global health burden, and molecular diagnostics such as Real-Time RT-PCR have revealed frequent co-infections. However, precise quantification of viral RNA remains challenging. Digital PCR (dPCR) offers absolute quantification without standard curves and may improve diagnostic accuracy. This study compares dPCR and Real-Time RT-PCR in detecting and quantifying influenza A, influenza B, respiratory syncytial virus (RSV), and SARS-CoV-2 during the 2023–2024 tripledemic. *Methods*: A total of 123 respiratory samples were analysed and stratified by cycle threshold (Ct) values into high, medium, and low viral load categories. Both dPCR and Real-Time RT-PCR were used to quantify and compare viral loads across these categories. *Results*: dPCR demonstrated superior accuracy, particularly for high viral loads of influenza A, influenza B, and SARS-CoV-2, and for medium loads of RSV. It showed greater consistency and precision than Real-Time RT-PCR, especially in quantifying intermediate viral levels. *Conclusions*: These findings highlight the potential of dPCR to enhance respiratory virus diagnostics and support a better understanding of co-infection dynamics. Nonetheless, its routine implementation is currently limited by higher costs and reduced automation compared to Real-Time RT-PCR.

## 1. Introduction

Respiratory infections [1], particularly those caused by influenza viruses A and B, respiratory syncytial virus (RSV), and severe acute respiratory syndrome coronavirus 2 (SARS-CoV-2), are among the leading causes of global morbidity and mortality [2]. These pathogens are responsible for seasonal epidemics [3], which can escalate into severe public health crises when multiple viruses co-circulate, as was observed during the 2023–2024 ‘tripledemic’ [4,5,6].

The COVID-19 pandemic has affected the seasonal circulation of many respiratory pathogens, particularly influenza and RSV, underscoring the public health risk of co-infection with other respiratory viruses [7,8,9]. The concurrent circulation of influenza A (H1N1 and H3N2), influenza B, RSV, and SARS-CoV-2 during the tripledemic placed considerable strain on healthcare systems worldwide. In Europe, surveillance data from the 2023–2024 season indicated a predominance of influenza A over influenza B, with notable regional variation in subtype prevalence, reflecting the dynamic interplay between viral evolution and public health interventions [10]. According to the ECDC Annual Epidemiological Report, the 2023–2024 influenza season in Europe was shorter than the previous year, with sentinel positivity exceeding the 10% epidemic threshold for 15 weeks (compared to 25 weeks in 2022–2023). The peak positivity rate reached 39% in week 52 of 2023, with influenza A(H1N1)pdm09 viruses predominating among hospitalised patients. RSV followed a typical seasonal pattern, contributing significantly to the burden on healthcare systems, particularly in paediatric and elderly populations. Similarly, in the United States, influenza and RSV hospitalisations during 2022–2023 reached unprecedented levels, exacerbating the challenges posed by resurgent SARS-CoV-2 infections [11,12]. This scenario underscores the necessity of robust diagnostic tools capable of accurately identifying and quantifying multiple pathogens simultaneously, thereby aiding in the management of such complex epidemics [13].

Recent studies have highlighted the unique epidemiological and clinical challenges posed by these viruses when they co-circulate. For instance, spatial–temporal analyses in the United States revealed distinct clustering patterns for the ‘tripledemic,’ with high-risk regions emerging in late 2022 and early 2023 [14]. The overlapping clinical presentations of these viruses, which include fever, cough, dyspnoea, and systemic manifestations, complicate diagnosis, emphasising the need for accurate and rapid molecular diagnostic techniques [15,16,17]. Moreover, the tripledemic demonstrated the importance of maintaining robust vaccination programmes [18].

Real-Time Reverse Transcription Polymerase Chain Reaction (RT-PCR) has long been the gold standard for detecting RNA viruses, including influenza A and B, RSV, and SARS-CoV-2. However, quantification via Real-Time RT-PCR depends on standard curves, which can introduce variability and limit precision, particularly in the presence of inhibitors [19,20]. This limitation becomes especially problematic in co-infection scenarios [21], where multiple viral genomes in a single sample, combined with complex respiratory matrices containing mucus and cellular debris, may differentially affect amplification efficiency, resulting in inconsistent Ct values and reduced reliability of quantification.

Digital PCR (dPCR) has emerged as a robust alternative, offering absolute quantification without the need for standard curves. By partitioning the PCR mixture into thousands of individual reactions, dPCR enables precise counting of target molecules, thereby improving sensitivity and reproducibility. Studies have shown that dPCR can detect low levels of viral RNA with greater consistency than Real-Time RT-PCR, particularly in samples with low viral load [22]. Among dPCR platforms, droplet digital PCR (ddPCR) and nanowell-based systems, such as the QIAcuity by QIAGEN, offer distinct methodologies for partitioning reactions. ddPCR generates thousands of nanolitre-sized droplets, each functioning as an independent PCR reaction vessel [23]. In contrast, the QIAcuity system employs fixed nanowells on a microfluidic chip, facilitating high-throughput processing and seamless integration with automated workflows. While both platforms offer comparable sensitivity and precision, the QIAcuity system allows for faster setup and reduced sample handling, making it well-suited to high-throughput laboratory environments [24,25].

Quantitative viral load data provide critical insights into the dynamics of infection, including disease severity, transmissibility, and treatment response. For instance, higher viral loads have been associated with increased risk of hospitalisation and worse clinical outcomes in both influenza and SARS-CoV-2 infections. In the context of RSV, viral load correlates with disease severity in paediatric and elderly populations, where early intervention is crucial. Moreover, accurate quantification is essential for monitoring antiviral efficacy, particularly in immunocompromised patients or those with prolonged viral shedding.

From a public health perspective, viral load measurements can inform infection control strategies, such as isolation duration and cohorting, especially during periods of co-circulation of multiple pathogens, as observed during the 2023–2024 tripledemic. Quantification also facilitates the identification of co-infections and the assessment of their relative contribution to disease burden, which is not possible with qualitative detection alone.

Given the global impact of respiratory infections and the critical need for robust diagnostic tools, this study aims to compare the performance of dPCR and Real-Time RT-PCR in detecting and quantifying influenza viruses A and B, RSV, and SARS-CoV-2. However, although the study explores the technical capability of dPCR to detect co-infections, the limited number of such cases in our dataset precluded a formal analysis of co-infection dynamics or clinical outcomes.

## 2. Materials and Methods

### 2.1. Study Design and Sample Collection

Respiratory samples were collected between November 2023 and April 2024 as part of routine surveillance conducted by the Laboratory of Microbiology and Virology in Bolzano, Italy. Samples were obtained from symptomatic patients presenting with respiratory symptoms, including fever, cough, dyspnoea, and other influenza-like illness (ILI) manifestations, in both inpatient and outpatient settings. Inclusion criteria required confirmed positivity for at least one of the target viruses, Influenza A, Influenza B, RSV, or SARS-CoV-2, based on Real-Time RT-PCR results. No additional selection criteria were applied, ensuring that the dataset reflects the diagnostic spectrum encountered during routine seasonal surveillance.

Respiratory samples are inherently heterogeneous due to variable mucus content, epithelial cell debris, and potential PCR inhibitors. These factors can affect nucleic acid extraction and amplification efficiency, particularly in Real-Time RT-PCR. Since dPCR partitions reactions into thousands of nanowells, it is less susceptible to such matrix effects, offering improved robustness in complex clinical specimens.

Samples were stratified into three groups according to Ct values: high (≤25), medium (25.1–30), and low (>30). These thresholds are used in clinical virology to estimate viral burden and inform infection control decisions. Lower Ct values typically indicate higher concentrations of viral RNA, whereas higher Ct values may reflect low levels of viral RNA.

A total of 122 nasopharyngeal swabs and one bronchoalveolar lavage (BAL) sample were analysed. The sample distribution included 28 Influenza A/H1N1, 12 Influenza A/H3N2, 18 Influenza B, 26 RSV, and 22 SARS-CoV-2 positive samples. Additionally, six samples showed RSV + Influenza coinfections, one sample showed RSV + SARS-CoV-2 codetection, and ten samples were included as negative controls.

All metadata, including patient demographics, was anonymised in accordance with established ethical guidelines.

### 2.2. Real-Time RT-PCR Workflow

Nucleic acid extraction was performed using the STARlet Seegene automated platform (Seegene, Seoul, Republic of Korea) in combination with the STARMag 96 X 4 Universal Cartridge Kit (Thermo Fisher Scientific, Waltham, MC, USA). Extracted RNA was subjected to multiplex Real-Time RT-PCR using commercial respiratory panel kits, Allplex Respiratory Panel 1A, 2, and 3 (Seegene, Seoul, Republic of Korea), targeting specific viral genes. Fluorescent signal detection and Ct value determination were carried out using a CFX96 thermocycler (Bio-Rad, Hercules, CA, USA). Internal controls were included in all the assays to ensure the quality of extraction and amplification.

### 2.3. Digital PCR Workflow

RNA extraction for dPCR was conducted using the KingFisher Flex system (Thermo Fisher Scientific, Waltham (MC), USA) with the MagMax Viral/Pathogen kit. dPCR assays were performed on the QIAcuity platform (Qiagen, Hilden, Germany) using a five-target multiplex format. Primer–probe mixes specific for Influenza A, Influenza B, RSV, SARS-CoV-2, and an internal control were optimised to minimise cross-reactivity. The primer–probe sets were part of a commercially validated kit and are not publicly disclosed. Optimisation involved empirical adjustments of primer and probe concentrations within the validated framework provided by the manufacturer to ensure optimal performance under laboratory conditions.

Samples were loaded into nanowell plates, partitioned into approximately 26,000 wells, and subjected to endpoint PCR. Fluorescent signals were detected and analysed using QIAcuity Suite software v.0.1, which calculated the absolute copy number of each target.

### 2.4. Statistical Analysis

Descriptive analysis of RNA concentration values was performed for both Real-Time RT-PCR and dPCR data. Measures of central tendency and variability were calculated. Data were evaluated separately by virus type (Influenza A, Influenza B, RSV, and SARS-CoV-2) and by session date.

Outliers in dPCR data were identified using boxplot visualisation, defined as values outside the interquartile range (IQR) multiplied by 1.5. The Kruskal–Wallis test was used to compare distributions of samples with low, medium, and high viral RNA concentrations, followed by post hoc pairwise comparisons.

The analysed samples were reclassified into two groups according to the Real-Time RT-PCR value: ≤25 (high viral RNA concentration) and >25 (low/medium viral RNA concentration). The ROC (Receiver Operating Characteristic) curve analysis was applied to these data to describe and compare the diagnostic accuracy of the dPCR technique. The ROC curve is a graphical representation that plots the number of true positive results (sensitivity) on the x-axis and the number of false negative results (1-specificity) on the y-axis, calculated for a range of cut-off values. A test with poor diagnostic efficiency is characterised by a curve that is close to the bisector, while a test with good performance is characterised by a curve that is shifted upwards towards the left corner. The ROC curve also allows the diagnostic test to be evaluated based on the value of the area under the curve. By comparing the combinations of sensitivity/specificity values defined by the ROC curve, it was possible to estimate the most appropriate threshold (optimal cut-off point) of dPCR.

ROC curve analyses were performed on the aggregated dataset, combining samples from all virus types. This approach was selected to identify general dPCR thresholds corresponding to Ct-based viral load categories. Virus-specific ROC analyses were not conducted due to the limited number of samples per virus type, particularly in the medium and low Ct ranges, which would have reduced statistical power and reliability.

All statistical analyses were performed using IBM SPSS software (version 20.0; Armonk, New York, NY, USA: IBM Corp.). A *p*-value < 0.05 (two-tailed test) was considered statistically significant.

## 3. Results

### 3.1. dPCR Assay Setup and Optimisation

A total of 12 dPCR sessions were conducted, each including a subset of the samples. Twenty-seven samples were repeated to optimise the dilution protocol. During the initial session, three dilution conditions (undiluted, 1:10, and 1:100) were tested to identify the most suitable approach for subsequent analyses. The results indicated that a 1:100 dilution was optimal for most samples. Samples with undetectable or inconclusive results at 1:100 dilution, typically those with Ct values above 30, were reanalysed using 1:10 and undiluted conditions. Appendix A provides detailed data on the distribution of analysed samples.

Quantification results obtained via dPCR are expressed as copies per microlitre (copies/µL) of extracted RNA eluate. No back-calculation or volume correction was applied to account for the original sample input volume. This approach reflects the analytical output of the QIAcuity system and ensures consistency across measurements.

### 3.2. Quantification of Viral RNA Concentration by dPCR

#### 3.2.1. Influenza A

Among the influenza A-positive samples, those with Ct values ranging from 18 to 25 showed viral RNA concentrations between 745.10 and 775,260.00 copies/µL when analysed using 1:10 and 1:100 dilutions. Samples with Ct values of 26 to 30 yielded viral RNA concentrations ranging from 5.42 to 1896.00 copies/µL, with measurements performed undiluted and at dilutions of 1:10 and 1:100. Finally, samples with Ct values of 31 to 39 produced viral RNA concentrations between 1.14 and 796.80 copies/µL across all tested dilutions (Appendix A).

#### 3.2.2. Influenza B

For Influenza B positive samples, those with Ct values ranging from 18 to 25 demonstrated viral RNA concentrations between 6471.00 and 559,970.00 copies/µL when diluted at 1:100. Samples with Ct values ranging from 26 to 30 exhibited viral RNA concentrations between 432.80 and 8104.00 copies/µL when analysed at 1:10 and 1:100 dilutions. Finally, samples with Ct values ranging from 31 to 39 yielded viral RNA concentrations between 0.39 and 226.10 copies/µL when tested undiluted and at a 1:10 dilution (Appendix A).

#### 3.2.3. Respiratory Syncytial Virus

Among the RSV samples, those with Ct values ranging from 17 to 25 exhibited viral RNA concentrations between 335.90 and 576,520.00 copies/µL, as measured using undiluted, 1:10, and 1:100 dilutions. Samples with Ct values of 26 to 30 showed viral RNA concentrations ranging from 581.00 to 37,860.00 copies/µL when analysed at dilutions of 1:10 and 1:100. For samples with Ct values between 31 and 42, viral RNA concentrations ranged from 2.61 to 144.50 copies/µL across all tested dilutions. One sample lacking a Ct value due to instrumentation error yielded 10.38 copies/µL when analysed at a 1:10 dilution (Appendix A).

#### 3.2.4. SARS-CoV-2

The SARS-CoV-2 samples included in the study ranged in Ct values from 11 to 25, indicative of viral RNA concentrations between 138.50 and 1,297,130.00 copies/µL. Three samples with Ct values of 26 to 27 yielded viral RNA concentrations ranging from 370.60 to 1210.00 copies/µL, while two samples with Ct values of 34 produced viral RNA concentrations of 2.16 and 5.50 copies/µL, respectively (Appendix A).

### 3.3. Comparison Between dPCR and Real-Time RT-PCR

Digital PCR demonstrated improved accuracy and consistency in quantifying viral RNA compared to Real-Time RT-PCR across all Ct categories.

Due to the lack of calibrators, Real-Time RT-PCR results could not be expressed in copies/µL. Instead, we stratified samples by Ct value and compared these categories with dPCR-derived copy numbers.

Sample classification into high, medium, and low viral material was based on Ct values from Real-Time RT-PCR, which served as the reference method. Although direct sample-by-sample comparison was not feasible due to differences in assay design and quantification principles, the distribution of dPCR-derived copy numbers across Ct-defined categories provides a meaningful framework for comparison. This reflects the clinical use of Ct values as a semi-quantitative reference and allows assessment of dPCR performance in terms of consistency and precision, as illustrated in Figure 1, Figure 2 and Figure 3. While alternative classification criteria, such as symptom severity, could also be informative, such data were not available in this study.

At high viral material levels (Ct ≤ 25), dPCR provided accurate quantification with minimal variability between replicates (Figure 1). These results closely matched expected distributions of RNA concentrations across Ct-defined categories, supporting the internal consistency of dPCR measurements.

At intermediate viral materials (Ct 25.1–30), dPCR maintained robust performance (Figure 1, Figure 2 and Figure 3; Appendix A). RSV samples (Figure 3), which showed higher variability in Real-Time RT-PCR measurements, were accurately quantified by dPCR.

A few samples showed co-infections, with combinations such as Influenza A-RSV and Influenza B-SARS-CoV-2 being the most common. dPCR enabled simultaneous quantification of multiple targets within these samples, providing insight into the relative contribution of each virus. No additional co-infections among the respiratory viruses examined were identified during the surveillance period beyond those explicitly reported.

Outliers observed in dPCR quantification likely reflect biological variability or differences in assay sensitivity rather than misclassification.

### 3.4. Receiver Operating Characteristic Curves’ Analysis

Using Real-Time RT-PCR Ct values as a reference, samples were categorised into three groups based on viral RNA concentration: (1) Ct ≤ 25, indicating high concentration; (2) 25 < Ct ≤ 30, indicating medium concentration; and (3) Ct > 30, indicating low concentration. dPCR exhibited significantly greater precision and accuracy for samples with high viral RNA concentration, while its performance for medium and low concentrations was comparable to Real-Time RT-PCR (Figure 1).

Subsequent analysis by viral type revealed that dPCR performed significantly better for Influenza A and B viruses (Figure 2 and Appendix A), particularly at high viral concentrations. Differences between dPCR and Real-Time RT-PCR were less pronounced for medium and low concentrations. For RSV (Figure 3), dPCR outperformed Real-Time RT-PCR in detecting subtle differences in medium and high viral RNA concentrations, while performance was comparable for low concentrations. For SARS-CoV-2 (Appendix A), dPCR showed high precision at high viral concentrations, although the limited number of positive samples may have introduced bias.

Receiver Operating Characteristic (ROC) curves were used to assess sensitivity and specificity across varying cut-off values (Figure 4 and Figure 5). The initial ROC analysis, distinguishing high (Ct ≤ 25) from medium-low (Ct > 25) concentrations, identified an optimal cut-off of 4000 copies/µL (Figure 4 and Appendix A), yielding 91.8% sensitivity, 80.5% specificity, and 87.25% overall accuracy. A second ROC analysis, differentiating low (Ct > 30) from medium (25 < Ct ≤ 30) concentrations, identified an optimal cut-off of 239.6 copies/µL (Figure 5 and Appendix A), with 84.0% sensitivity, 93.8% specificity, and 87.80% accuracy.

The reported sensitivity and specificity values refer to dPCR’s ability to stratify samples by viral RNA concentration, not to detect infection status. All samples included in the ROC analysis were confirmed positive by Real-Time RT-PCR.

The number of samples in some Ct-defined categories, particularly for low and medium viral loads, was limited. This may affect the statistical power of comparisons and the robustness of ROC-derived thresholds.

## 4. Discussion

This study compared two molecular diagnostic approaches: Real-Time RT-PCR, the current gold standard for RNA virus detection, and digital PCR (dPCR), an emerging technology offering absolute quantification of targets with higher sensitivity and precision. Compared to Real-Time RT-PCR, dPCR offers several advantages: it enables absolute quantification without standard curves, a known limitation of Real-Time RT-PCR, is less affected by inhibitors, provides higher precision and reproducibility, especially at low and intermediate viral loads, and allows accurate multiplexing with minimal target competition. These features make dPCR particularly suitable for complex clinical samples and for applications requiring precise viral load monitoring. dPCR is particularly advantageous in multi-target detection scenarios where accurate quantification is essential for clinical decision-making.

The primary objective of the study was to assess the sensitivity of both methods and evaluate the feasibility of integrating dPCR into routine diagnostics to improve the accuracy of viral RNA quantification.

For samples with high viral RNA material (Ct ≤ 25), dPCR demonstrated superior sensitivity and precision across all viruses analysed. For instance, dPCR quantified up to 775,260 copies/µL for Influenza A and up to 1,297,130 copies/µL for SARS-CoV-2. In contrast, Real-Time RT-PCR, which relies on Ct values, exhibited greater variability, particularly at high viral concentrations. The improved performance of dPCR is reflected in the tighter distribution and lower variability of measurements, as shown in Figure 1 and Figure 2, especially in the high viral load category. At medium viral RNA material (Ct 25–30), dPCR maintained robust performance, particularly for RSV, where Real-Time RT-PCR showed increased variability. Although the differences between the two methods were less pronounced in samples with low viral RNA material (Ct > 30), dPCR still provided consistent detection and quantification, whereas Real-Time RT-PCR performance declined. These findings are consistent with previous studies, such as Gentilini et al. [26], which demonstrated that dPCR provides more accurate quantification of SARS-CoV-2 RNA, particularly in samples with low viral burden.

Unlike Ct values, which provide only relative estimates, dPCR directly measures RNA copy numbers, reducing dependence on amplification efficiency and standard curves.

A small number of samples exceeded the validated dynamic range of the QIAcuity platform; these were retained to reflect the full spectrum of viral RNA concentrations observed during the 2023–2024 season. However, results from these samples should be interpreted with caution due to potential saturation effects that may compromise quantification accuracy.

Of particular note, dPCR exhibited superior sensitivity for RSV even at intermediate viral material (Ct 25–30), a critical observation given the clinical implications of RSV in paediatric and elderly populations. As noted by other authors [27], accurate detection and quantification of respiratory pathogens are essential for guiding treatment decisions. dPCR’s ability to detect intermediate viral loads may help prevent clinical mismanagement, such as unnecessary escalation of therapy, and better inform clinicians in assessing patient risk. In clinical practice, precise quantification may also support decisions regarding isolation duration, initiation of antiviral therapy, or escalation of care, particularly in immunocompromised individuals or those with persistent symptoms.

dPCR is valuable for monitoring viral dynamics over time, especially in conditions where viral reservoirs persist at low levels [28]. In scenarios where viral material is low (Ct > 30), dPCR consistently detected and quantified the target, outperforming Real-Time RT-PCR, which exhibited reduced performance. This aligns with findings from Gentilini et al. [26], who showed that dPCR improves the sensitivity of SARS-CoV-2 RNA quantification, even in cases of low viral presence. Their findings support the use of dPCR in clinical settings where precise measurements are required for accurate diagnosis and therapeutic guidance.

The enhanced sensitivity of dPCR for low viral material samples offers an additional advantage, ensuring reliable detection and quantification, even in challenging clinical scenarios, which is essential for monitoring viral dynamics over time, particularly in patients undergoing antiviral therapies [24]. This is particularly relevant for early-stage infections, post-treatment monitoring, or immunocompromised patients, where conventional Real-Time RT-PCR may fail to detect or accurately quantify low-level targets. The added value of dPCR becomes more evident in low-copy contexts, where its partition-based approach ensures greater analytical sensitivity. Moreover, although longitudinal sampling was not included in this study, the ability of dPCR to detect subtle variations in viral RNA concentration suggests its potential utility in tracking infection dynamics over time.

However, in the case of this study, because the Real-Time RT-PCR data were obtained from routine diagnostic workflows optimised for qualitative detection, and dPCR was applied retrospectively using a multiplex quantification assay, direct sensitivity comparisons should be approached with caution.

Another significant advantage of dPCR lies in its ability to discriminate between multiple pathogens within the same sample. While multiplexing is also feasible with Real-Time RT-PCR, dPCR offers enhanced precision due to its partitioned format, which reduces competition between targets and allows more accurate quantification of each. This capability has been widely documented in the context of respiratory infections. For example, a recent study [27] showed that multiplex dPCR assays are highly effective in detecting and quantifying co-infections such as SARS-CoV-2 and RSV in the same clinical sample. The efficacy of dPCR in detecting viral co-infections, such as RSV and SARS-CoV-2, has been well established.

While a limited number of samples in this investigation showed co-infections (for example, RSV with influenza A or SARS-CoV-2), the dataset was insufficient to support a statistically significant comparison between dPCR and Real-Time RT-PCR in this context. Therefore, whereas dPCR may give advantages in co-infection circumstances, this was not decisively established in the current investigation. However, dPCR demonstrated the technical capability to simultaneously and independently quantify multiple viral targets within the same reaction, without the need for standard curves or concern for target competition.

Receiver operating characteristic (ROC) curve analysis further validated the superior performance of dPCR, establishing cut-off values that reliably distinguished between high and low viral materials. ROC analysis enables the determination of optimal thresholds for viral copies per microlitre, balancing sensitivity and specificity to maximise diagnostic accuracy. For instance, a cut-off of 4000 copies/µL demonstrated an accuracy of 87.25% for high viral materials, with excellent sensitivity (91.8%) but moderate specificity (80.5%). Conversely, for low viral materials (Ct > 30), a cut-off of 239.6 copies/µL achieved higher specificity (93.8%) but slightly lower sensitivity (84.0%).

ROC analyses were based on pooled data across all viruses to derive general cut-off values. This methodological choice was necessary due to the limited sample size per virus type and reflects the study’s aim to evaluate dPCR performance across a representative spectrum of respiratory infections. Future studies should investigate virus-specific threshold optimisation using larger and more balanced datasets.

This study has several limitations that should be considered when interpreting the results. First, the number of SARS-CoV-2 positive samples was limited, which may affect the generalisability of the findings for this virus. Additionally, the number of samples in some Ct-defined categories, particularly for low and medium viral loads, was small, potentially reducing the statistical power of comparisons and the robustness of ROC-derived thresholds. Outliers observed in dPCR quantification likely reflect biological variability or differences in assay sensitivity rather than misclassification. Future studies with larger and more balanced sample groups are warranted to confirm these observations.

Another limitation of this study is represented by the absence of normalisation to the number of host cells in each sample. While relative quantification using cellular reference genes (e.g., RNase P) is recommended to account for sampling variability, this was not feasible in our retrospective design. The Real-Time RT-PCR data used in this study were derived from routine diagnostic workflows, where assays are validated for qualitative detection and semi-quantitative interpretation based on Ct values. These assays are not routinely calibrated for absolute quantification, and commercial calibrators were not available for all viral targets. Consequently, Ct values were used as a proxy for viral load, consistent with clinical practice. While this approach does not allow direct comparison in copies/µL, it reflects real-world diagnostic conditions and supports the relevance of dPCR as a complementary quantification tool. Future studies should consider incorporating such internal controls to enable standardised viral load comparisons.

It is also important to acknowledge that different RNA extraction platforms were used for Real-Time RT-PCR and dPCR workflows, namely, the STARlet Seegene system and the KingFisher Flex system, respectively. While this may introduce variability in RNA yield and purity, both extraction systems are CE-IVD certified and widely validated for clinical diagnostics. The aim of this study was to evaluate each detection method within its optimised diagnostic workflow, reflecting real-world laboratory conditions.

Despite the clear diagnostic advantages of dPCR, its integration into routine clinical workflows is currently limited by several challenges. These include operational costs, assay complexity, and longer turnaround times compared to real-time RT-PCR, which limit its current applicability. The adoption of dPCR in high-throughput clinical diagnostics requires automation and cost-effective solutions to overcome these barriers [27]. Compared to Real-Time RT-PCR, which typically delivers results within 1.5–3 hours and is compatible with high-throughput workflows, dPCR involves additional steps such as sample dilution and partitioning, increasing hands-on time. Instrumentation and reagents for dPCR are also more expensive. These factors currently restrict its use to specialised applications, although ongoing technological advances may improve its feasibility for routine diagnostics. The advancement of dPCR workflows, simplified and cost-effective, is therefore critical to facilitating their wider adoption.

Future research should explore the broader clinical applications of dPCR, including its role in monitoring antiviral treatment efficacy, evaluating co-infection dynamics, and detecting emerging viral pathogens [26,29]. The ability to detect and quantify low-level viral material and accurately monitor co-infections positions dPCR as a valuable tool in managing complex viral diseases, particularly in the context of global health threats like pandemics and emerging infectious diseases.

Finally, a critical barrier to the clinical adoption of dPCR is the lack of CE-IVD certification for many platforms and assays. Currently, many dPCR systems, including the QIAcuity platform and the kits used in this study, are marketed as Research Use Only (RUO) and are not CE-IVD certified. As such, they cannot be used for patient diagnostics unless validated by the laboratories.

## 5. Conclusions

dPCR has demonstrated substantial improvements in the detection and quantification of respiratory viruses compared to Real-Time RT-PCR, particularly in clinical scenarios requiring precise viral material measurements. However, to fully realise the clinical potential of dPCR, further efforts are required to standardise protocols, validate assays under appropriate regulatory frameworks, and improve cost-efficiency. Additionally, larger-scale studies incorporating clinical metadata and longitudinal sampling are needed to clarify the role of dPCR in patient management and epidemiological surveillance. Despite current operational challenges, the integration of dPCR into clinical diagnostics has the potential to transform respiratory virus management and improve patient outcomes.

## Figures and Tables

**Figure 1 viruses-17-01259-f001:**
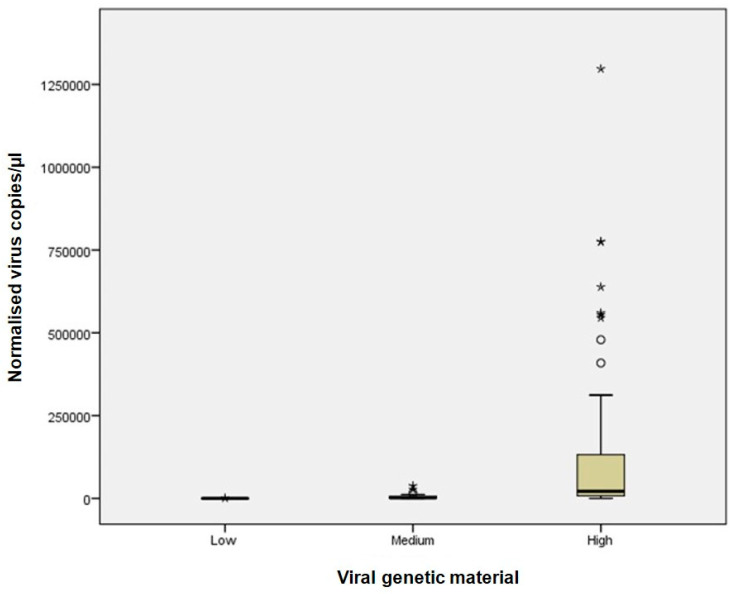
Comparison of dPCR and Real-Time RT-PCR results for all analysed samples. The box plot illustrates the performance of dPCR versus Real-Time RT-PCR across three viral RNA concentration categories: high (Ct ≤ 25), medium (25 < Ct ≤ 30), and low (Ct > 30). Outliers excluded from the analysis are marked with asterisks. Open circles represent individual data points included in statistical analysis, after they have been checked.

**Figure 2 viruses-17-01259-f002:**
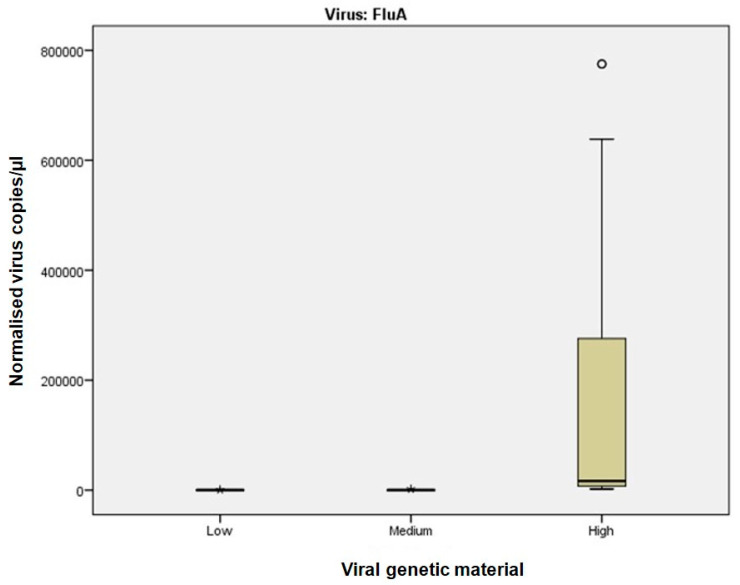
dPCR versus Real-Time RT-PCR for Influenza A virus samples stratified by Ct values. This figure highlights the performance of dPCR in quantifying viral RNA concentration for Influenza A virus samples. Precision was higher in samples with high viral RNA concentration (Ct ≤ 25), while differences diminished in medium and low categories. Open circles represent individual data points included in statistical analysis, after they have been checked. Asterisks (*) indicate samples excluded from analysis due to outlier status.

**Figure 3 viruses-17-01259-f003:**
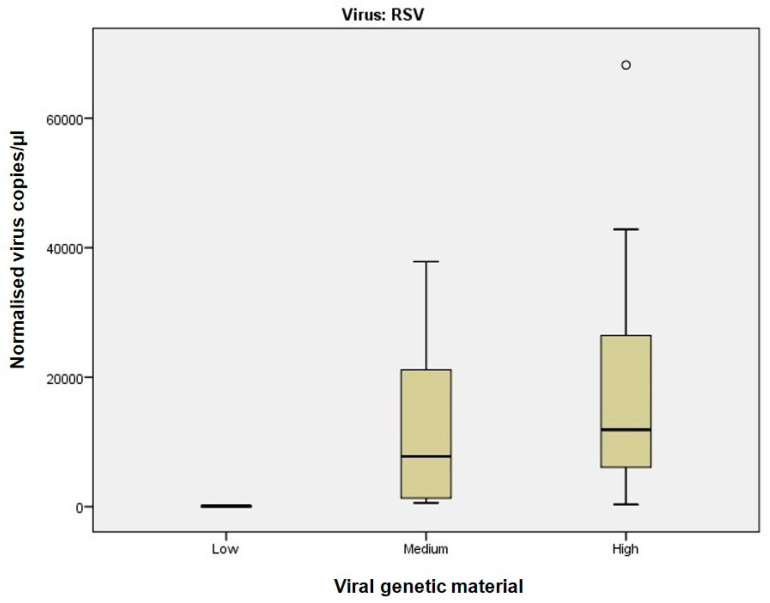
dPCR versus Real-Time RT-PCR for RSV samples stratified by Ct values. The figure demonstrates the improved sensitivity of dPCR in detecting minimal differences for RSV samples with medium and high viral RNA concentration, while showing comparable performance to Real-Time RT-PCR in low viral load samples. Open circles represent individual data points included in statistical analysis, after they have been checked.

**Figure 4 viruses-17-01259-f004:**
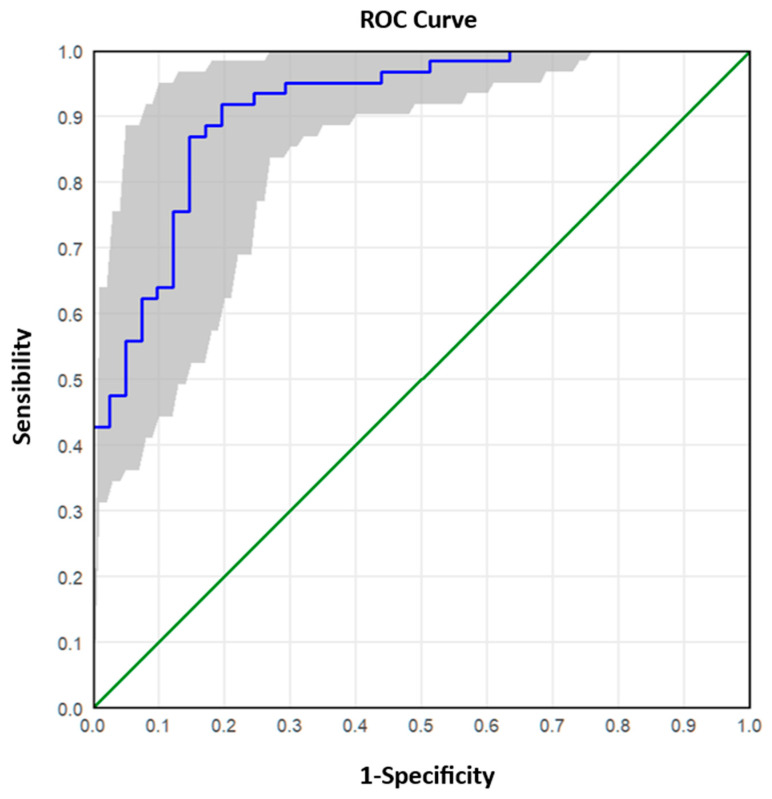
ROC curve for the discrimination between high and medium–low viral RNA concentrations. The grey band around the Roc curve represents the 95% confidence interval. The area under the curve (AUC) is equal to 0.914. The ROC curve determines the optimal cut-off at 4000 copies/μL, balancing sensitivity (91.8%) and specificity (80.5%).

**Figure 5 viruses-17-01259-f005:**
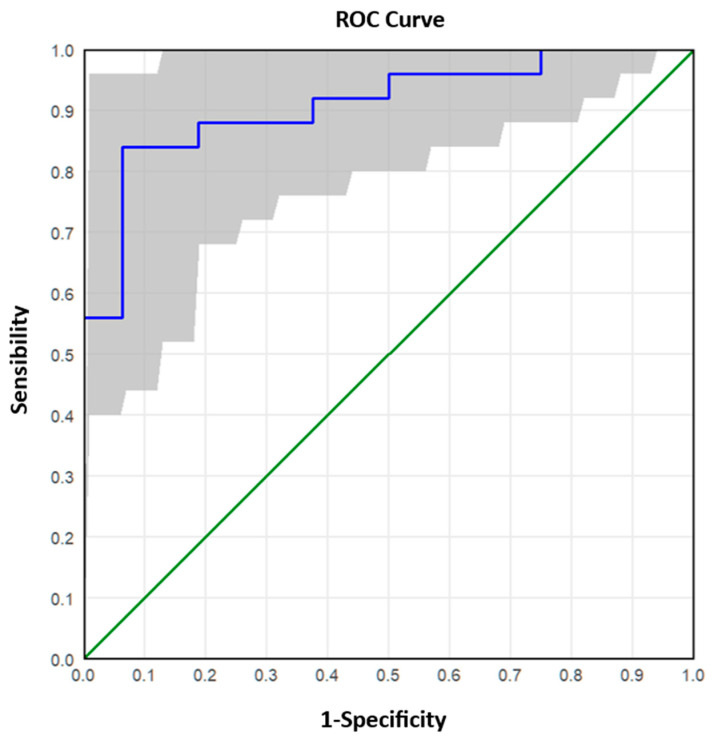
ROC curve for the discrimination between low and medium viral RNA concentrations. The grey band around the Roc curve represents the 95% confidence interval. The area under the curve (AUC) is equal to 0.910. The curve shows an optimal cut-off at 239.6 copies/μL with sensitivity (84.0%) and specificity (93.8%).

## Data Availability

All data generated or analysed during this study are included in this published article [and its Appendix A].

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
