# Peer review of "Comparative Performance of Digital PCR and Real-Time RT-PCR in Respiratory Virus Diagnostics"

_viruses, 2025, doi:10.3390/v17091259_

Round 1
Reviewer 1 Report
Comments and Suggestions for Authors
Authors compare digital PCR and real-time PCR for the detection and quantification of influenza virus A/B, RSV, and SARS-CoV-2 with specimens collected during the 2023-2024 tripledemic season. By analyzing and stratifying the Ct values of real-time PCR and the RNA copy numbers of digital PCR, this study demonstrated a correlation between these two methods. Some major and minor points should be addressed.
major points:
Consider addressing the methodological consistency across the analyses. Figures 2 and 3 suggest different sensitivity patterns between FluA and RSV (particularly in the 'Medium' category), yet the subsequent ROC analyses in Figures 4 and 5 appear to treat the data uniformly. A brief discussion of how virus-specific differences were handled in the cutoff determination would strengthen the methodology section.
minor points:
In the Materials and Methods section, the criteria for collecting samples for routine surveillance should be clarified. This would strengthen the epidemiological content of the manuscript.
Supplementary Table 1 may be confusing. Consider the following improvements: (1) restructure the table with specimens in rows and viruses in columns for better understanding of co-infections, (2) include the threshold for "positive" in real-time PCR (Ct <40 ?) in the main text or legend, (3) provide positive control data, (4) standardize "Sars-CoV-2" as "SARS-CoV-2" for consistency with the main text.
In Figure 1, both open circles and asterisks are shown. Could you clarify whether the excluded samples correspond only to those marked with asterisks?
(Same problems exist in Figs 2 and 3.)
In Figures 4 and 5, the ROC analysis could be enhanced by considering the following points: (1) It would be helpful to include confidence intervals and AUC values for the ROC curves; (2) Please clarify which virus types (influenza A/B, RSV, SARS-CoV-2 or all of them ?) are being analyzed; (3) It would be useful to specify whether all positive virus samples were analyzed together or separately; (4) Please explain how sample #108 (RSV positive but RT-PCR Ct value is "undetermined" in Supplementary Table 1) was handled in the analysis; and (5) Consider defining the criteria used for categorizing RT-PCR results as positive versus negative when calculating sensitivity and specificity (e.g., whether "high" Ct values were considered positive and "medium-low" values as negative).
Regarding Section 3.5, if this section is to remain as a standalone section, I think it would be better to provide a specific figure or table.
Author Response
Reviewer 1
Comments and Suggestions for Authors
Authors compare digital PCR and real-time PCR for the detection and quantification of influenza virus A/B, RSV, and SARS-CoV-2 with specimens collected during the 2023-2024 tripledemic season. By analyzing and stratifying the Ct values of real-time PCR and the RNA copy numbers of digital PCR, this study demonstrated a correlation between these two methods. Some major and minor points should be addressed.
major points:
- Consider addressing the methodological consistency across the analyses. Figures 2 and 3 suggest different sensitivity patterns between FluA and RSV (particularly in the 'Medium' category), yet the subsequent ROC analyses in Figures 4 and 5 appear to treat the data uniformly. A brief discussion of how virus-specific differences were handled in the cutoff determination would strengthen the methodology section.
The ROC analyses presented in Figures 4 and 5 were performed on the entire dataset, aggregating samples across all virus types. Due to the limited number of samples per virus type, especially in the medium and low Ct categories, we decided to pool the data, instead of performing virus-specific ROC analyses, as this would have compromised statistical robustness.
The pooling-approach was chosen to identify general cut-off thresholds for dPCR quantification that correspond to clinically relevant Ct categories (i.e., high, medium, and low viral load), as defined by Real-Time RT-PCR.
While recognising that virus-specific optimisation may be warranted in future studies with larger cohorts, this methodological choice reflects the practical constraints of the dataset and the study’s aim to evaluate dPCR performance under real-world diagnostic conditions.
We have added a paragraph in the Materials and Methods (a) and Discussion (b) sections to clarify this methodological choice and its implications.
a: “ROC curve analyses were performed on the aggregated dataset, combining samples from all virus types. This approach was selected to identify general dPCR thresholds corresponding to Ct-based viral load categories. Virus-specific ROC analyses were not conducted due to the limited number of samples per virus type, particularly in the medium and low Ct ranges, which would have reduced statistical power and reliability.”
b: “ROC analyses were based on pooled data across all viruses to derive general cut-off values. This methodological choice was necessary due to the limited sample size per virus type and reflects the study’s aim to evaluate dPCR performance across a representative spectrum of respiratory infections. Future studies should investigate virus-specific thresh-old optimisation using larger and more balanced datasets.”
minor points:
- In the Materials and Methods section, the criteria for collecting samples for routine surveillance should be clarified. This would strengthen the epidemiological content of the manuscript.
We have now clarified that respiratory samples were collected as part of routine diagnostic surveillance conducted by the Laboratory of Microbiology and Virology in Bolzano, Italy, during the 2023–2024 respiratory virus season. Samples were obtained from symptomatic patients presenting with respiratory symptoms, including fever, cough, dyspnoea, and other influenza-like illness (ILI) manifestations, in both inpatient and outpatient settings. Inclusion criteria required laboratory-confirmed positivity for at least one of the target viruses based on Real-Time RT-PCR results. The study did not involve pre-selection based on clinical severity or demographic characteristics, thereby reflecting the real-world diagnostic spectrum encountered during seasonal surveillance.
“Samples were obtained from symptomatic patients presenting with respiratory symp-toms, including fever, cough, dyspnoea, and other influenza-like illness (ILI) manifesta-tions, in both inpatient and outpatient settings. Inclusion criteria required confirmed posi-tivity for at least one of the target viruses, Influenza A, Influenza B, RSV, or SARS-CoV-2, based on Real-Time RT-PCR results. No additional selection criteria were applied, ensuring that the dataset reflects the diagnostic spectrum encountered during routine seasonal surveillance.”
- Supplementary Table 1 may be confusing. Consider the following improvements: (1) restructure the table with specimens in rows and viruses in columns for better understanding of co-infections, (2) include the threshold for "positive" in real-time PCR (Ct <40 ?) in the main text or legend, (3) provide positive control data, (4) standardize "SARS-CoV-2 " as "SARS-CoV-2 " for consistency with the main text.
We have revised the table.
- In Figure 1, both open circles and asterisks are shown. Could you clarify whether the excluded samples correspond only to those marked with asterisks? (Same problems exist in Figs 2 and 3.)
Only the samples marked with asterisks (*) were excluded from the statistical analyses. These samples were identified as outliers and were not included in the statistics. Open circles represent individual data points that were included in the analysis. We have revised the figure legends to clarify this distinction.
“Open circles represent individual data points included in statistical analysis, after they have been checked. Asterisks (*) indicate samples excluded from analysis due to outlier status.”
- In Figures 4 and 5, the ROC analysis could be enhanced by considering the following points: (1) It would be helpful to include confidence intervals and AUC values for the ROC curves; (2) Please clarify which virus types (influenza A/B, RSV, SARS-CoV-2 or all of them ?) are being analyzed; (3) It would be useful to specify whether all positive virus samples were analyzed together or separately; (4) Please explain how sample #108 (RSV positive but RT-PCR Ct value is "undetermined" in Supplementary Table 1) was handled in the analysis; and (5) Consider defining the criteria used for categorizing RT-PCR results as positive versus negative when calculating sensitivity and specificity (e.g., whether "high" Ct values were considered positive and "medium-low" values as negative).
(1) It would be helpful to include confidence intervals and AUC values for the ROC curves
Figure 4. ROC curve for the discrimination between high and medium-low viral RNA concentrations. The grey band around the Roc curve represents the 95% confidence interval. The area under the curve (AUC) is equal to 0.914. The ROC curve determines the optimal cut-off at 4,000 copies/μL, balancing sensitivity (91.8%) and specificity (80.5%).
Figure 5. ROC curve for the discrimination between low and medium viral RNA concentrations. The grey band around the Roc curve represents the 95% confidence interval. The area under the curve (AUC) is equal to 0.910. The curve shows an optimal cut-off at 239.6 copies/μL with sensitivity (84.0%) and specificity (93.8%).
(2) Please clarify which virus types (influenza A/B, RSV, SARS-CoV-2 or all of them ?) are being analyzed
ROC curve analyses were performed on data from all virus types.
(3) It would be useful to specify whether all positive virus samples were analyzed together or separately;
All positive virus samples were analyzed together
(4) Please explain how sample #108 (RSV positive but RT-PCR Ct value is "undetermined" in Supplementary Table 1) was handled in the analysis;
Sample #108 (RSV positive but Real Time RT-PCR Ct value is "undetermined" in Supplementary Table 1) was excluded by the analysis.
5) Consider defining the criteria used for categorizing RT-PCR results as positive versus negative when calculating sensitivity and specificity (e.g., whether "high" Ct values were considered positive and "medium-low" values as negative).
For each ROC curve, once the optimal cut-off value (c) of copies/µL has been identified, cases are classified as positive if value ≥ ? and negative if value < c.
- Regarding Section 3.5, if this section is to remain as a standalone section, I think it would be better to provide a specific figure or table.
Section 3.5 addresses co-infection analysis, which is indeed relevant in the context of respiratory virus diagnostics. However, we acknowledge that the number of co-infected samples in our dataset was very limited (n=7), and therefore insufficient to support a standalone figure or table with meaningful statistical or visual value.
We have now incorporated Section 3.5 into Section 3.3.

Reviewer 2 Report
Comments and Suggestions for Authors
The authors present a comparative study of the quantification of the viral load of four respiratory viruses, either by real-time RT-PCR, which is the gold standard, or by digital PCR.
The article is generally well written and easy to understand.
In terms of scientific content, the authors do not explain in the introduction why it is important or necessary to quantify the viral load of these four respiratory viruses. In addition, the sampling methods for respiratory infections are very heterogeneous, which increases the complexity of these samples, in addition to the presence of mucus, etc. The advantages of dPCR over real-time RT-PCR should be better argued. Several authors also suggest standardisation by relative quantification based on the number of cells present in the sample. This standardisation, even if only an estimate, could be added to the study. Quantification in copies/µl using the real-time RT-PCR technique could also be added using calibrators. A direct comparison of the viral load in copies/µl using the two techniques (rather than a comparison of Ct values on the one hand and viral load in copies/µl on the other) could better show the differences in quantification and thus the difficulties in estimating the viral load.
The figures would be clearer if a semi-logarithmic scale were used; furthermore, they should be inserted into the body of the text.
The choice of Ct value cut-offs should be justified. The same applies to the values chosen for supplementary tables S2 and S3 (values for which virus?).
Write SARS-CoV-2 (not Sars-CoV-2) throughout (including figures).
I disagree with the authors: dPCR is not more sensitive than RT-PCR. In supplementary table S1, only one sample (No. 108) is detected by dPCR and not by RT-PCR. This point needs to be reviewed. Furthermore, the superiority of dPCR in cases of co-infection is not clearly demonstrated in this study.
Author Response
Reviewer 2
Comments and Suggestions for Authors
The authors present a comparative study of the quantification of the viral load of four respiratory viruses, either by real-time RT-PCR, which is the gold standard, or by digital PCR.
The article is generally well written and easy to understand.
- In terms of scientific content, the authors do not explain in the introduction why it is important or necessary to quantify the viral load of these four respiratory viruses.
We have revised the Introduction to explicitly articulate the clinical and epidemiological relevance of quantifying viral load for influenza A, influenza B, respiratory syncytial virus (RSV), and SARS-CoV-2 .
“Quantitative viral load data provide critical insights into the dynamics of infection, including disease severity, transmissibility, and treatment response. For instance, higher viral loads have been associated with increased risk of hospitalisation and worse clinical outcomes in both influenza and SARS-CoV-2 infections (Branche et al., 2025; Brandolini et al., 2021). In the context of RSV, viral load correlates with disease severity in paediatric and elderly populations, where early intervention is crucial (Tang et al., 2024). Moreover, accurate quantification is essential for monitoring antiviral efficacy, particularly in immunocompromised patients or those with prolonged viral shedding.
From a public health perspective, viral load measurements can inform infection control strategies, such as isolation duration and cohorting, especially during periods of co-circulation of multiple pathogens, as observed during the 2023–2024 tripledemic. Quantification also facilitates the identification of co-infections and the assessment of their relative contribution to disease burden, which is not possible with qualitative detection alone.”
- In addition, the sampling methods for respiratory infections are very heterogeneous, which increases the complexity of these samples, in addition to the presence of mucus, etc.
We agree that respiratory samples are inherently heterogeneous due to variable mucus content, epithelial cells, and potential PCR inhibitors. We have added a sentence in the Materials and Methods section to acknowledge this complexity and to highlight that dPCR, thanks to the partition-based design, is less susceptible to such inhibitory effects. This reinforces the rationale for evaluating dPCR as a potentially more robust quantification method in complex clinical matrices.
“Respiratory samples are inherently heterogeneous due to variable mucus content, epithelial cell debris, and potential PCR inhibitors. These factors can affect nucleic acid extraction and amplification efficiency, particularly in Real-Time RT-PCR. dPCR, by partitioning reactions into thousands of nanowells, is less susceptible to such matrix effects, offering improved robustness in complex clinical specimens.”
- The advantages of dPCR over real-time RT-PCR should be better argued.
We have expanded the Discussion to provide a more comprehensive comparison between dPCR and real-time RT-PCR.
“Compared to Real-Time RT-PCR, dPCR offers several advantages: it enables absolute quantification without standard curves, a known limitation of Real-Time RT-PCR, is less affected by inhibitors, provides higher precision and reproducibility, especially at low and intermediate viral loads, and allows accurate multiplexing with minimal target competi-tion. These features make dPCR particularly suitable for complex clinical samples and for applications requiring precise viral load monitoring.”
- Several authors also suggest standardisation by relative quantification based on the number of cells present in the sample. This standardisation, even if only an estimate, could be added to the study.
We recognise the significance of normalising viral load to the number of host cells, since this helps account for variability in sample collection and increase comparability between specimens. However, because this was a retrospective investigation using regular diagnostic samples, cell quantification was not feasible. We have included a paragraph in the Discussion to acknowledge this.
“Another limitation of this study is represented by the absence of normalisation to the number of host cells in each sample. While relative quantification using cellular reference genes (e.g., RNase P) is recommended to account for sampling variability, this was not feasible in our retrospective design. The Real-Time RT-PCR data used in this study were derived from routine diagnostic workflows, where assays are validated for qualitative de-tection and semi-quantitative interpretation based on Ct values. These assays are not rou-tinely calibrated for absolute quantification, and commercial calibrators were not availa-ble for all viral targets. Consequently, Ct values were used as a proxy for viral load, con-sistent with clinical practice. While this approach does not allow direct comparison in copies/µl, it reflects real-world diagnostic conditions and supports the relevance of dPCR as a complementary quantification tool.Future studies should consider incorporating such internal controls to enable standardised viral load comparisons.”
- Quantification in copies/µl using the real-time RT-PCR technique could also be added using calibrators.
We agree that quantification using Real-time RT-PCR would require the use of external calibrators or standard curves derived from known RNA concentrations. However, the Real-Time RT-PCR data used in this study were obtained from routine diagnostic workflows, where assays are validated for qualitative detection and semi-quantitative interpretation based on cycle threshold (Ct) values. These assays are not routinely calibrated for absolute quantification, and calibrators were not available for all viral targets at the time of analysis. In clinical practice, Ct values are widely used as a proxy for viral load, providing a relative measure of RNA concentration that informs patient management and infection control decisions. We have clarified this point in the revised manuscript and acknowledged the limitation that real-time RT-PCR results could not be expressed in copies/µl.
“Another limitation of this study is represented by the absence of normalisation to the number of host cells in each sample. While relative quantification using cellular reference genes (e.g., RNase P) is recommended to account for sampling variability, this was not feasible in our retrospective design. The Real-Time RT-PCR data used in this study were derived from routine diagnostic workflows, where assays are validated for qualitative de-tection and semi-quantitative interpretation based on Ct values. These assays are not rou-tinely calibrated for absolute quantification, and commercial calibrators were not availa-ble for all viral targets. Consequently, Ct values were used as a proxy for viral load, con-sistent with clinical practice. While this approach does not allow direct comparison in copies/µl, it reflects real-world diagnostic conditions and supports the relevance of dPCR as a complementary quantification tool.Future studies should consider incorporating such internal controls to enable standardised viral load comparisons.”
- A direct comparison of the viral load in copies/µl using the two techniques (rather than a comparison of Ct values on the one hand and viral load in copies/µl on the other) could better show the differences in quantification and thus the difficulties in estimating the viral load.
While a direct comparison in copies/µl between the two methods would be ideal, as stated above, it was not feasible in our study. Instead, we stratified samples by Ct value and compared these categories with dPCR-derived copy numbers. We included a statement with the results.
“Due to the lack of calibrators, Real-Time RT-PCR results could not be expressed in copies/µl. Instead, we stratified samples by Ct value and compared these categories with dPCR-derived copy numbers.”
- The figures would be clearer if a semi-logarithmic scale were used; furthermore, they should be inserted into the body of the text.
We chose the linear scale to maintain absolute proportionality between values and ensure consistency with the statistical analyses presented. The linear scale also allows the reader to immediately interpret the observed values in absolute terms.
The position of figures in the manuscript is determined by the journal's formatting rules rather than the authors' preferences.
- The choice of Ct value cut-offs should be justified.
The Ct cut-offs used in this study, ≤25 for high viral RNA, 25.1–30 for medium, and >30 for low, were selected based on established clinical and diagnostic conventions. These thresholds and categories are routinely used in diagnostic reporting. In our study, they served as a reference framework for comparing the performance of dPCR across different viral load strata. We have added a justification in the Materials and Methods section.
“These thresholds are used in clinical virology to estimate viral burden and inform infection control decisions. Lower Ct values typically indicate higher concentrations of viral RNA, whereas higher Ct values may reflect low-level of viral RNA.”
- The same applies to the values chosen for supplementary tables S2 and S3 (values for which virus?)
These values were derived from ROC curve analyses across all viruses combined and, along with the thresholds of 4,000 and 239.6 copies/µl, represent optimal cut-offs for distinguishing high vs. medium/low and medium vs. low viral loads, respectively, and are not virus-specific.
- Write SARS-CoV-2 (not SARS-CoV-2 ) throughout (including figures).
We amended the manuscript.
- I disagree with the authors: dPCR is not more sensitive than RT-PCR. In supplementary table S1, only one sample (No. 108) is detected by dPCR and not by RT-PCR. This point needs to be reviewed.
We have revised the Discussion to clarify that our study does not demonstrate superior analytical sensitivity of dPCR in terms of detection. Our study primarily focused on the comparative quantification performance of dPCR rather than its diagnostic sensitivity. The Real-Time RT-PCR data were obtained from routine clinical diagnostics, where assays are optimised for high-throughput qualitative detection. In contrast, dPCR was applied retrospectively to the same samples using a research-use multiplex assay designed for absolute quantification. Our findings shows that dPCR provides more consistent and precise quantification across a wide range of viral loads.
“However, in the case of this study, because the Real-Time RT-PCR data were obtained from routine diagnostic workflows optimised for qualitative detection, and dPCR was ap-plied retrospectively using a multiplex quantification assay, direct sensitivity compari-sons should be approached with caution.”
- Furthermore, the superiority of dPCR in cases of co-infection is not clearly demonstrated in this study.
Due to the limited number of co-infected samples (n=7), we were unable to perform a robust statistical analysis. While dPCR enables simultaneous quantification of multiple targets, our dataset does not permit definitive conclusions regarding its superiority in co-infection scenarios. Nonetheless, we believe it is important to highlight the technical capability of dPCR to simultaneously quantify multiple viral targets within a single reaction. Unlike real-time RT-PCR, which may suffer from competition between targets in multiplex assays and relies on relative quantification, dPCR partitions each reaction into thousands of nanowells, allowing for independent and absolute quantification of each target. This feature is particularly advantageous in co-infection scenarios, where accurate assessment of the relative viral burden may inform clinical decision-making.
We have revised the Discussion to acknowledge this limitation.
“While a limited number of samples in this investigation showed co-infections (for example, RSV with influenza A or SARS-CoV-2), the dataset was insufficient to support a statistically significant comparison between dPCR and Real-Time RT-PCR in this context. Therefore, whereas dPCR may give advantages in co-infection circumstances, this was not decisively established in the current investigation. However, dPCR demonstrated the technical capability to simultaneously and independently quantify multiple viral targets within the same reaction, without the need for standard curves or concern for target com-petition.”

Round 2
Reviewer 1 Report
Comments and Suggestions for Authors
Dear authors,
Thank you for your detailed response, which has greatly improved my understanding.
Particularly, the revised Table 1 is much clearer. However, I understand that co-infection cases may be limited in number, but the current Table 1 does not reflect the co-infection numbers mentioned in the Materials and Methods section. It would be helpful to include this data.
Author Response
Comment: Particularly, the revised Table 1 is much clearer. However, I understand that co-infection cases may be limited in number, but the current Table 1 does not reflect the co-infection numbers mentioned in the Materials and Methods section. It would be helpful to include this data.
Response: We have revised the table.
Reviewer 2 Report
Comments and Suggestions for Authors
The authors have corrected their manuscript as requested, and the conclusions are more balanced.
Author Response
We thank you the Reviewer.